# Experimental evidence of tetrahedral symmetry breaking in SiO$_2$ glass under pressure

Yoshio Kono [1✉], Koji Ohara [2], Nozomi M. Kondo[1], Hiroki Yamada[2], Satoshi Hiroi[2], Fumiya Noritake[3], Kiyofumi Nitta[2], Oki Sekizawa[2], Yuji Higo[2], Yoshinori Tange [2], Hirokatsu Yumoto [2,4], Takahisa Koyama[2,4], Hiroshi Yamazaki[2,4], Yasunori Senba[2,4], Haruhiko Ohashi[2,4], Shunji Goto[2,4], Ichiro Inoue[4], Yujiro Hayashi[4], Kenji Tamasaku[4], Taito Osaka[4], Jumpei Yamada[4] & Makina Yabashi [4]

Bimodal behavior in the translational order of silicon's second shell in SiO$_2$ liquid at high temperatures and high pressures has been recognized in theoretical studies, and the fraction of the S state with high tetrahedrality is considered as structural origin of the anomalous properties. However, it has not been well identified in experiment. Here we show experimental evidence of a bimodal behavior in the translational order of silicon's second shell in SiO$_2$ glass under pressure. SiO$_2$ glass shows tetrahedral symmetry structure with separation between the first and second shells of silicon at low pressures, which corresponds to the S state structure reported in SiO$_2$ liquid. On the other hand, at high pressures, the silicon's second shell collapses onto the first shell, and more silicon atoms locate in the first shell. These observations indicate breaking of local tetrahedral symmetry in SiO$_2$ glass under pressure, as well as SiO$_2$ liquid.

[1] Geodynamics Research Center, Ehime University, 2-5 Bunkyo-cho, Matsuyama 790-8577, Japan. [2] Japan Synchrotron Radiation Research Institute, 1-1-1 Kouto, Sayo-cho, Sayo-gun, Hyogo 679-5198, Japan. [3] Graduate Faculty of Interdisciplinary Research, University of Yamanashi, 4-3-11 Takeda, Kofu, Yamanashi 400-8511, Japan. [4] RIKEN SPring-8 Center, 1-1-1 Kouto, Sayo-cho, Sayo-gun, Hyogo 679-5148, Japan. ✉email: kono.yoshio.rj@ehime-u.ac.jp

Understanding the structural origin of the anomalous properties of tetrahedral liquids and amorphous materials at high-temperature and high-pressure conditions is of great interest in a wide range of scientific fields such as physics, chemistry, geoscience, and materials science. The water anomaly is the most famous example. Water shows a density maximum at high temperatures and a compressibility maximum at high pressures[1–5]. Similar to water, $SiO_2$ liquid shows anomalous density and compressibility behaviors at high temperatures and high pressures[4,6–8]. In addition, it has been known that $SiO_2$ glass also shows compressibility maximum (bulk modulus minimum) at high pressures of ~2–3 GPa under room temperature condition[9–11]. Since $SiO_2$ is ubiquitous on the Earth, understanding the $SiO_2$'s anomaly is fundamental not only in physics, but also in geophysics to understand nature of silicate magmas in the Earth and planet, and in materials science as a prototype network-forming glass.

Theoretical studies of $SiO_2$ liquid[4,7] suggest that the second shell structure of silicon is the key to understand the anomalous properties of $SiO_2$ liquid at high temperatures and high pressures. Structural parameters $g_5$ (the average density of the fifth neighbor of silicon atom)[7] and $z$ ($z = \delta_{ji} - \delta_{j'i}$, where $\delta_{ji}$ and $\delta_{j'i}$ is the distance from each silicon atom i to the fifth nearest silicon neighbor j and to the fourth nearest oxygen neighbor j')[4] were developed to investigate the second shell structure in $SiO_2$ liquid. The theoretical studies reported a bimodal distribution in these structural parameters with varying temperature[4,7], and suggest the relevance of the two-state model description to $SiO_2$ liquid. The S and $\rho$ states are assigned to the high and low distributions in the parameter $z$, respectively[4]. The low-density S state in $SiO_2$ liquid consists of four silicon neighbor atoms in the first shell and exhibits high tetrahedral order with a clear separation between the first and second shell. On the other hand, the $\rho$ state has more silicon neighbor atoms in the first shell and shows lower tetrahedral order than the S state. The fraction of the S state with high tetrahedrality is considered to be the controlling parameter of the anomalous properties of $SiO_2$ liquid at high temperatures and high pressures in theoretical study[4]. However, there has been no experimental observation of the structure of the silicon's second shell in $SiO_2$ liquid and/or glass at high temperature and/or high-pressure conditions.

Although the experimental investigation of the structure of $SiO_2$ liquid at extremely high-temperature conditions has been still challenging, there have been several experimental studies for the structure of $SiO_2$ glass at high pressures under room temperature conditions. However, structural investigations of the previous high-pressure experimental studies were limited mainly to the nearest neighbor Si-O distance and coordination number[12–15], and/or fast sharp diffraction peak of the structure factor [S(Q)][13,16], due to experimental difficulties of structural analysis beyond the nearest neighbor distance at in situ high-pressure conditions. Although some recent studies have investigated S(Q) of $SiO_2$ glass at a high Q range to ~17 Å$^{-1}$ by using multi-angle energy-dispersive X-ray diffraction technique[15,17], energy-dispersive X-ray diffraction measurement using white X-ray inevitably contains uncertainty in X-ray intensity profile as a function of energy and resultantly Q. Therefore, it was difficult to conduct detailed structural analysis in the previous studies[15,17]. On the other hand, a recent ambient pressure study has enabled precise structural analysis of $SiO_2$ glass beyond the nearest neighbor distances by utilizing high-energy X-ray and neutron diffraction measurements combined with the MD (molecular dynamics simulation)-RMC (reverse Monte Carlo) modeling[18]. Such detailed structural analysis may open a new way to find the possible existence of the bimodal feature in the second shell structure of silicon in $SiO_2$ glass. However, structural analysis of

$SiO_2$ glass samples recovered from high-pressure and high-temperature syntheses shows only a slight change in the translational order in the second shells of silicon[18].

In this work, we carry out in situ high-pressure pair distribution function measurement of $SiO_2$ glass by utilizing high-flux and high-energy X-rays from undulator sources at BL37XU and BL05XU beamlines in SPring-8 (see Methods). By combining the high-pressure experimental S(Q) precisely determined by using monochromatic X-ray at a wide range of Q up to 19–20 Å$^{-1}$ with the MD-RMC modeling, we are able to investigate the detailed structural behavior of $SiO_2$ glass beyond the nearest neighbor distances under in situ high-pressure conditions. We find bimodal features in the translational order of the silicon's second shell in terms of the structural parameter $z$ and in the void radius formed from silicon atoms in $SiO_2$ glass under pressure. The bimodal behavior in the distribution of the parameter $z$ observed in $SiO_2$ glass with varying pressure in this study is consistent with that simulated in $SiO_2$ liquid with varying temperature in the theoretical study[4]. The structure of $SiO_2$ glass with the characteristic distribution of the parameter $z$ at 2.4–2.7 Å shows a tetrahedral symmetry structure formed from the nearest four silicon atoms in the first shell, and the first and second shells are clearly separated as the fifth neighbor silicon atom locates in the second shell. The structural feature corresponds to the low-density S state structure reported in the theoretical study of $SiO_2$ liquid[4]. On the other hand, the structure of $SiO_2$ glass with the characteristic distribution of $z$ at 1.7 Å shows that the fifth neighbor silicon atom locates in the first shell, which indicates the collapse of the silicon's second shell onto the first shell and breaking of local tetrahedral symmetry in $SiO_2$ glass under pressure, as well as theoretical observation in $SiO_2$ liquid at high temperatures and high pressures[4].

## Results

**Structure factor and pair distribution function**. Figure 1 shows S(Q) of $SiO_2$ glass determined at in situ high-pressure conditions up to 6.0 GPa. The S(Q) of $SiO_2$ glass shows distinct changes at the pressure ranges between 2.3 and 4.4 GPa (Fig. 1). Above 4.4 GPa, a new second peak clearly starts to appear at ~2.9 Å$^{-1}$ (Fig. 1b). In addition, the third peak in S(Q) at ~5 Å$^{-1}$ shows a shoulder peak at the low Q side below 2.3 GPa, while it becomes a single broad peak above 4.4 GPa. The structure of $SiO_2$ glass at high-pressure conditions are analyzed by the MD-RMC modeling based on the experimentally observed S(Q) (Fig. 1a). Supplementary Fig. 1 shows partial structure factors and partial pair distribution functions of Si-O [$S_{SiO}(Q)$ and $g_{SiO}(r)$], O-O [$S_{OO}(Q)$ and $g_{OO}(r)$], and Si-Si [$S_{SiSi}(Q)$ and $g_{SiSi}(r)$], respectively, obtained by the MD-RMC modeling. Partial pair distribution functions of $g_{SiO}(r)$ (Supplementary Fig. 1d) and $g_{OO}(r)$ (Supplementary Fig. 1e) show no distinct change with varying pressure, indicating that the first neighbor Si-O and O-O structures stay the same up to 6.0 GPa. In contrast, there are marked changes in $g_{SiSi}(r)$ at high pressures (Supplementary Fig. 1f). The first neighbor Si-Si distance shortens with increasing pressure, indicating a decrease in Si-O-Si angle, which is consistent with the previous Raman spectroscopy observation[19]. In addition, we find a marked change in the second peak of $g_{SiSi}(r)$ at ~4–6 Å. The second shell structure of silicon is considered to be a key to understand the microscopic structural behavior of $SiO_2$ liquid in theoretical studies[4,7].

**Translational order of silicon's second shell**. Ref. 4 proposed a structural parameter $z$ to measure the translational order of the silicon's second shell in terms of the relative distance between the first and second shells of silicon. The parameter $z$ is defined as

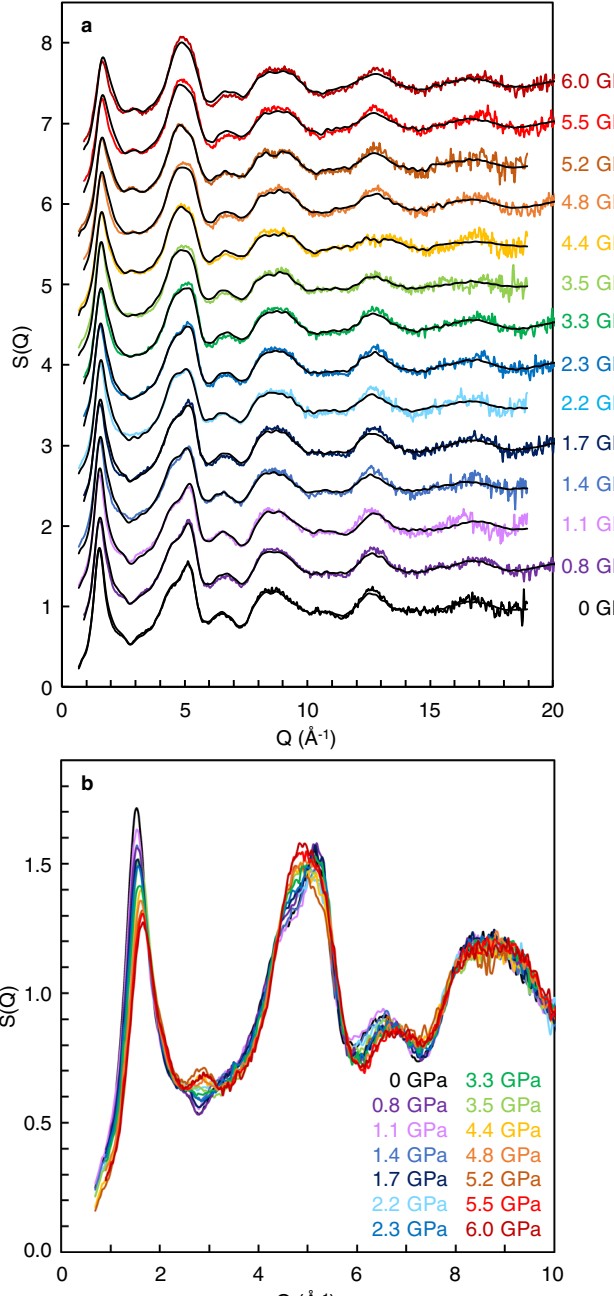

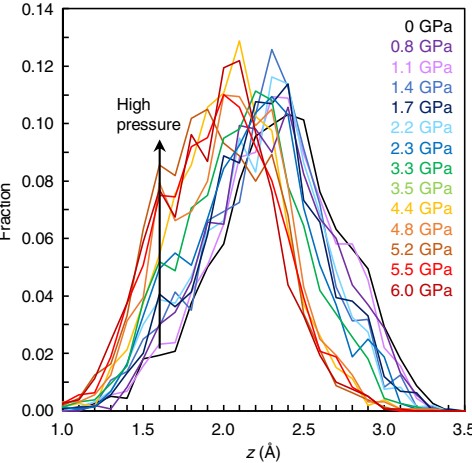

**Fig. 2 Translational order in SiO₂ glass as a function of the parameter z with varying pressure.** The parameter $z$ is defined as $z = \delta_{ji} - \delta_{j'i}$, where $\delta_{ji}$ and $\delta_{j'i}$ is the distance from each silicon atom i to the fifth nearest silicon neighbor j and to the fourth nearest oxygen neighbor j', respectively[4]. Source data are provided as a Source Data file.

**Fig. 1 Structure factor [S(Q)] of SiO₂ glass at high pressures. a** Color lines represent experimentally observed S(Q), and black lines show S(Q) of the MD-RMC structure model derived based on the experimentally observed S(Q) at each pressure condition. S(Q) is displayed by a vertical offset of +0.5 with increasing pressure. **b** Enlarged view of the experimentally observed S(Q) at the Q range between 0 and 10 Å⁻¹. S(Q) is displayed without vertical offset. There are clear changes in the oscillations at ~2.9 and ~5 Å⁻¹ with increasing pressure.

$z = \delta_{ji} - \delta_{j'i}$, where $\delta_{ji}$ and $\delta_{j'i}$ is the distance from each silicon atom i to the fifth nearest silicon neighbor j and to the fourth nearest oxygen neighbor j', respectively. Figure 2 shows the translational order in SiO₂ glass under in situ high-pressure conditions, analyzed from our observed MD-RMC model structure. We find a bimodal feature in the distribution of the parameter $z$ with varying pressure. There is a single distribution below 2.3 GPa, while another distribution at the $z = 1.6$–1.7 Å starts to increase rapidly with increasing pressure above 2.3 GPa. The

behavior of the distribution of the parameter $z$ observed in SiO₂ glass with varying pressure in this study is consistent with that simulated in SiO₂ liquid with varying temperature[4].

It is important to note that the change of the distribution of the parameter $z$ at 1.6–1.7 Å (Fig. 2) closely correlates with the evolution of the second peak at ~2.9 Å⁻¹ in the experimentally observed S(Q) (Fig. 1b). Supplementary Fig. 2 shows a simulated result of the influence of the second peak of S(Q) at ~2.9 Å⁻¹ on the distribution of the parameter $z$. The final MD-RMC model reproduces the experimentally observed S(Q) of SiO₂ glass at 5.2 GPa well including the second peak at ~2.9 Å⁻¹ (solid black line in Supplementary Fig. 2a). On the other hand, a simulated MD-RMC model shown by a broken black line reproduces most of the oscillation features of the experimentally observed S(Q), while it does not reproduce the second peak at ~2.9 Å⁻¹ (Supplementary Fig. 2a). The difference in the second peak of S(Q) at ~2.9 Å⁻¹ between two MD-RMC models yields a marked difference in the distribution of the parameter $z$ at 1.6–1.7 Å (Supplementary Fig. 2b). The final MD-RMC model shows a high fraction of the distribution of the parameter $z$ at 1.6–1.7 Å (solid black line), while the broken black line shows a markedly low fraction at $z = 1.6$–1.7 Å. The data indicate that the evolution of the second peak of S(Q) at ~2.9 Å⁻¹ closely correlates with the evolution of the distribution of the parameter $z$ at 1.6–1.7 Å. Therefore, we consider that the evolution of the second peak at ~2.9 Å⁻¹ in the experimentally observed S(Q) (Fig. 1b) represents the change in the translational order of the silicon's second shell in SiO₂ glass.

**Distributions of void radius formed from silicon atoms.** In order to understand distributions of silicon in SiO₂ glass under pressure, we conducted void radius analysis for silicon atoms based on Delaunay tessellation[20]. Only silicon atoms were used in the tetrahedralization, and a circumscribed sphere of the tetrahedron is defined as a void. The results show a distinct difference in the void radius distributions formed from silicon atoms with varying pressure (Fig. 3). Void radius above 4.4 GPa shows a single distribution with the peak position at ~3.1 Å, and there is almost no distribution above 3.6 Å (Fig. 3c). In contrast, the void radius distribution below 2.2 GPa shows another distribution of void radius at ~3.7–3.8 Å (Fig. 3a), in addition to the distribution

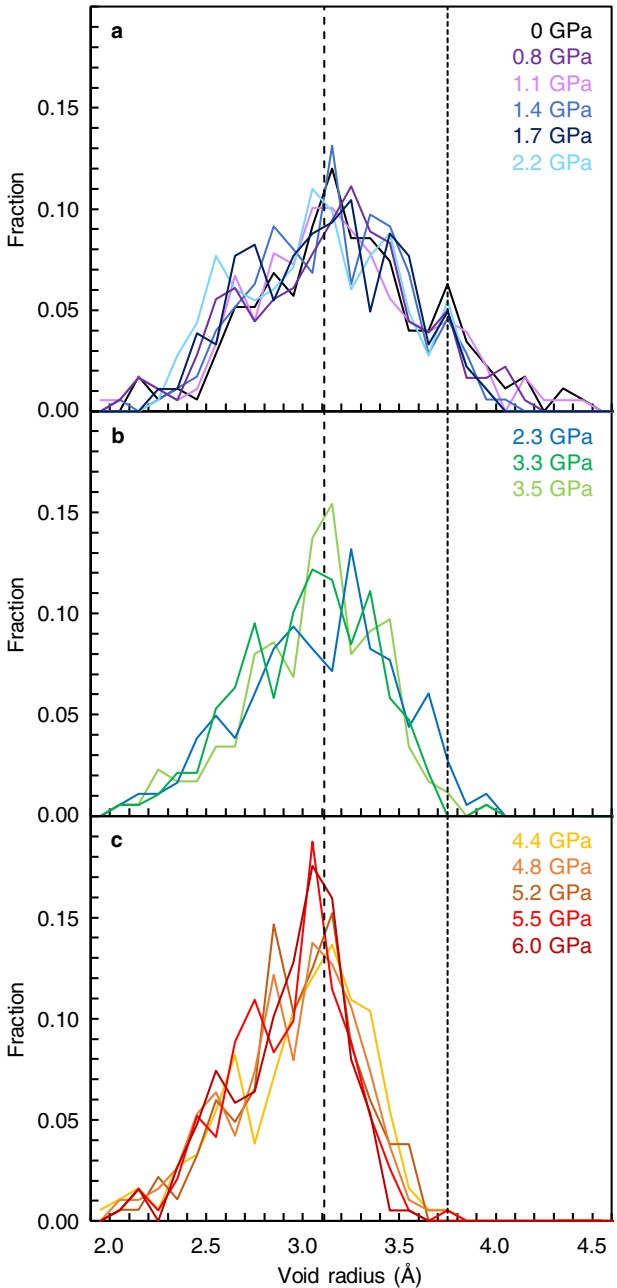

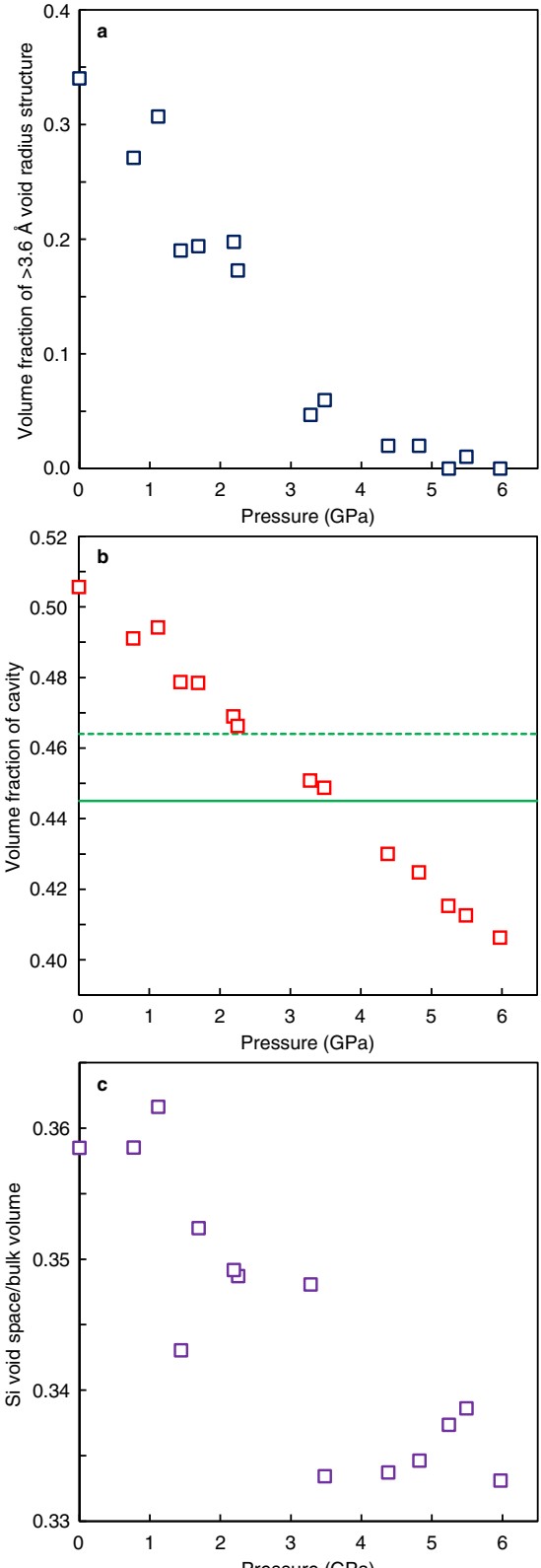

**Fig. 3 Distributions of void radius formed from silicon atoms in SiO₂ glass at high pressures. a** 0–2.2 GPa. **b** 2.3–3.5 GPa. **c** 4.4–6.0 GPa. The broken line at the void radius of 3.11 Å represents the ring size of a six-membered ring structure. The dotted line at the void radius of 3.75 Å is a guide for the eye. Source data are provided as a Source Data file.

at ~3.1–3.2 Å. The distribution at ~3.7–3.8 Å gradually disappears with increasing pressure between 2.3 and 3.5 GPa (Fig. 3b).

## Discussion

Our results show bimodal features in the void radius formed from silicon atoms (Fig. 3) and in the translational order of the silicon's second shell (Fig. 2). Both structural parameters characterize the distribution of silicon in SiO₂ glass. We observe two distributions in the void radius formed from silicon atoms below 2.2 GPa (Fig. 3a). We note that the characteristic void radius of ~3.1 Å in SiO₂ glass (Fig. 3) corresponds to the ring size of a six-membered ring structure (3.11 Å based on a calculation of the radius of gyration ($R_G$) for planar ring[21] by $R_G = \frac{1}{2}L\csc(\pi/K)$, where L is

the Si-Si distance (3.11 Å) and K is the ring size of 6). In fact, ring size statistics analysis in our SiO₂ glass also shows the mean ring size of ~6 at 0–6.0 GPa (Supplementary Fig. 3). These data indicate that SiO₂ glass is composed of a six-membered ring structure throughout the entire pressure conditions of this study between 0 and 6.0 GPa. On the other hand, there is another characteristic void radius distribution at ~3.7–3.8 Å in SiO₂ glass

**Fig. 4 Variation of a void structure formed from silicon atoms and volume fraction of cavity at high pressures. a** Volume fraction of >3.6 Å void radius structure at high pressures. **b** Volume fraction of cavity in SiO$_2$ glass at high pressures. Green lines represent the volume fraction of the cavity of 0.445 (solid green line) and 0.464 (broken green line), corresponding to the random loose packing limit of equal spheres (0.555 (ref. [22]) and 0.536 (ref. [23]), respectively. **c** Change of void space formed from silicon atoms (Si void space) relative to the change of the bulk volume of SiO$_2$ glass at high pressures. Source data are provided as a Source Data file.

below 2.2 GPa (Fig. 3a), which suggests the presence of another large void structure below 2.2 GPa.

The fraction of the large void structure in SiO$_2$ glass with varying pressure is calculated by defining the void radius larger than 3.6 Å as the >3.6 Å void radius structure (Fig. 4a). It is interesting to note that the threshold value of 3.6 Å in our SiO$_2$ glass is consistent with the value used to identify the low-density and high-density structures in terms of the structural parameter $g_5$ in SiO$_2$ liquid in a simulation study[7], which implies the relevance of the void radius analysis in this study to the second shell structures of silicon. The fraction of the >3.6 Å void radius structure decreases with increasing pressure, and it almost disappears above 3.3 GPa (Fig. 4a). We note that, at the same pressure condition as the disappearance of the >3.6 Å void radius structure, the volume fraction of cavity in SiO$_2$ glass reaches that of the random loose packing condition of equal spheres (Fig. 4b). The random loose packing of equal spheres is considered to be 0.555 (ref. [22]) and 0.536 (ref. [23]), which corresponds to the fraction of cavity of 0.445 and 0.464, respectively. The volume fraction of the cavity in SiO$_2$ glass increases at lower pressures, and it reaches that of the random loose packing limit at ~2.5–3.5 GPa (Fig. 4b). The random loose packing limit is considered to be the lowest limit of mechanically stable random packing arrangement of equal spheres, although the packing condition looser than the random loose packing limit is possible if spheres are strongly bonded as covalent bonding. Therefore, the cavity volume higher than the random loose packing limit in SiO$_2$ glass below ~2.5–3.5 GPa (Fig. 4b) implies a mechanically unstable structure. In fact, we find that void space formed from silicon atoms (Si void space) anomalously shrinks more than the change of the bulk volume of SiO$_2$ glass below ~2.3–3.5 GPa (Fig. 4c), which may be related to the anomalous properties of SiO$_2$ glass at pressures below ~2–3 GPa (refs. [9–11]). On the other hand, above 3.5 GPa, Si void space changes equally to the bulk volume change of SiO$_2$ glass (Fig. 4c), and SiO$_2$ glass exhibits normal behavior as the elastic properties start to increase with increasing pressure[9–11]. These data indicate that anomalous properties of SiO$_2$ glass is related to the change in the distribution of silicon atoms under pressure.

It has been known in theoretical studies of SiO$_2$ liquid[4,7] that the second shell structure of silicon is the key to understand the anomalous properties of SiO$_2$ liquid at high temperatures and high pressures. Theoretical study of SiO$_2$ liquid[4] showed a bimodal feature in the structural parameter $z$ with varying temperatures and suggested the relevance of a two-state description for SiO$_2$ liquid. SiO$_2$ liquid shows a single distribution of the parameter $z$ at ~2.4–2.5 Å at low temperatures, while another distribution at $z = $~1.6 Å increases with decreasing the distribution of $z = $~2.4–2.5 Å at high temperatures. The theoretical study of SiO$_2$ liquid[4] assigned the S and ρ states to the high and low distributions in the parameter $z$, respectively, and showed that the fraction of the low-density S state is the controlling parameter of the anomalous density behavior of SiO$_2$ liquid not only as a function of temperature at ambient pressure but also as a function of pressure at 5000 K.

Similar to the theoretical study of SiO$_2$ liquid[4], we observed a bimodal feature in the parameter $z$ in SiO$_2$ glass under pressure

(Fig. 2), which implies the presence of two structural features in SiO$_2$ glass as well SiO$_2$ liquid. In order to clarify the two structural features in SiO$_2$ glass, we conducted MD simulations using two potential models (BKS[24] and MSD[25] models) at 0 and 5.2 GPa (Fig. 5 and Supplementary Fig. 4). The BKS model has been used in the theoretical studies of SiO$_2$ liquid[4,7]. Both BKS and MSD models at 0 GPa show a single distribution at the $z = 2.4$ Å (BKS model) and $z = 2.7$ Å (MSD model), which are consistent with our result by experiment combined with MD-RMC modeling (Exp+MD-RMC) at 0 GPa (Fig. 5a). The partial pair distribution functions of Si-Si [$g_{SiSi}(r)$] at 0 GPa show a clear separation between the first and second shells of silicon distributions in both experiment and MD simulations (Fig. 5b). The structural feature corresponds to the low-density S state structure reported in the theoretical study of SiO$_2$ liquid[4]. The structure of SiO$_2$ glass with the distribution of $z$ at 2.4–2.7 Å at 0 GPa shows a tetrahedral symmetry structure formed from the nearest four silicon atoms in the first shell, and the first and second shells are clearly separated as the fifth neighbor silicon atom locates in the second shell at a farther distance than 4.1 Å (Fig. 5d, e).

On the other hand, our Exp + MD-RMC result at 5.2 GPa shows a marked increase in the distribution of $z$ at ~1.6–1.7 Å accompanied by a decrease in the distribution of $z$ at ~2.4 Å (Fig. 5a). In MD simulations, although the MSD model shows almost single distribution at $z = 2.4$ Å with a subtle increase in the distribution of $z$ at 1.5–1.6 Å, the BKS model shows a clear bimodal feature in the distribution of $z$ (Fig. 5a). The BKS model at 5.2 GPa shows a significant increase in the distribution of $z$ at 1.7 Å accompanied by a decrease in the distribution of $z$ at 2.4 Å, which is consistent with our Exp + MD-RMC result. The $g_{SiSi}(r)$ of our Exp + MD-RMC model and BKS model at 5.2 GPa show a marked increase of the intensity at ~3.3–4.0 Å, and the separation between the first and second shells of silicon distribution disappears (Fig. 5b). The structure of SiO$_2$ glass with the distribution of $z$ at 1.7 Å in the BKS model at 5.2 GPa shows that the fifth neighbor silicon atom locates in the first shell, which indicates the collapse of the second shell onto the first shell and breaking of local tetrahedral symmetry (Fig. 5c).

It is important to note that the two structural features obtained in SiO$_2$ glass under pressure in this study correspond to the S and ρ state structures of SiO$_2$ liquid reported in the theoretical study[4]. For SiO$_2$ liquid, the low-density S state has four silicon neighbor atoms in the first shell and exhibits high tetrahedral order with a clear separation between the first and second shell[4]. On the other hand, the high-density ρ state of SiO$_2$ liquid shows more than four silicon atoms in the first shell by collapsing the second shell onto the first shell and exhibits low tetrahedral order. The structural features of the S and ρ state of SiO$_2$ liquid correspond to the two structural features obtained in SiO$_2$ glass with the characteristic distribution of $z$ at 2.4–2.7 Å (Fig. 5d, e) and 1.7 Å (Fig. 5c), respectively. These observations indicate the similarity of the structural behavior in SiO$_2$ glass under pressure obtained in this study to that theoretically simulated in SiO$_2$ liquid at high temperatures and high pressures[4]. SiO$_2$ glass mainly consists of the low-density S state with tetrahedral symmetry structure at low pressures. On the other hand, the local tetrahedral symmetry structure breaks at high pressures, and the fraction of the low-density S state in SiO$_2$ glass decreases under pressure, as well as theoretical observation in SiO$_2$ liquid at high temperatures and high pressures[4].

## Methods
**High-pressure experiment**. High-pressure experiments were conducted by using Paris-Edinburgh (PE) press. We used a cup-shaped PE anvil with a 12 mm cup diameter and a 3 mm flat bottom, and a standard PE cell assembly, which is mainly composed of a ZrO$_2$ cap, boron-epoxy gasket, and MgO ring[26]. SiO$_2$ glass sample

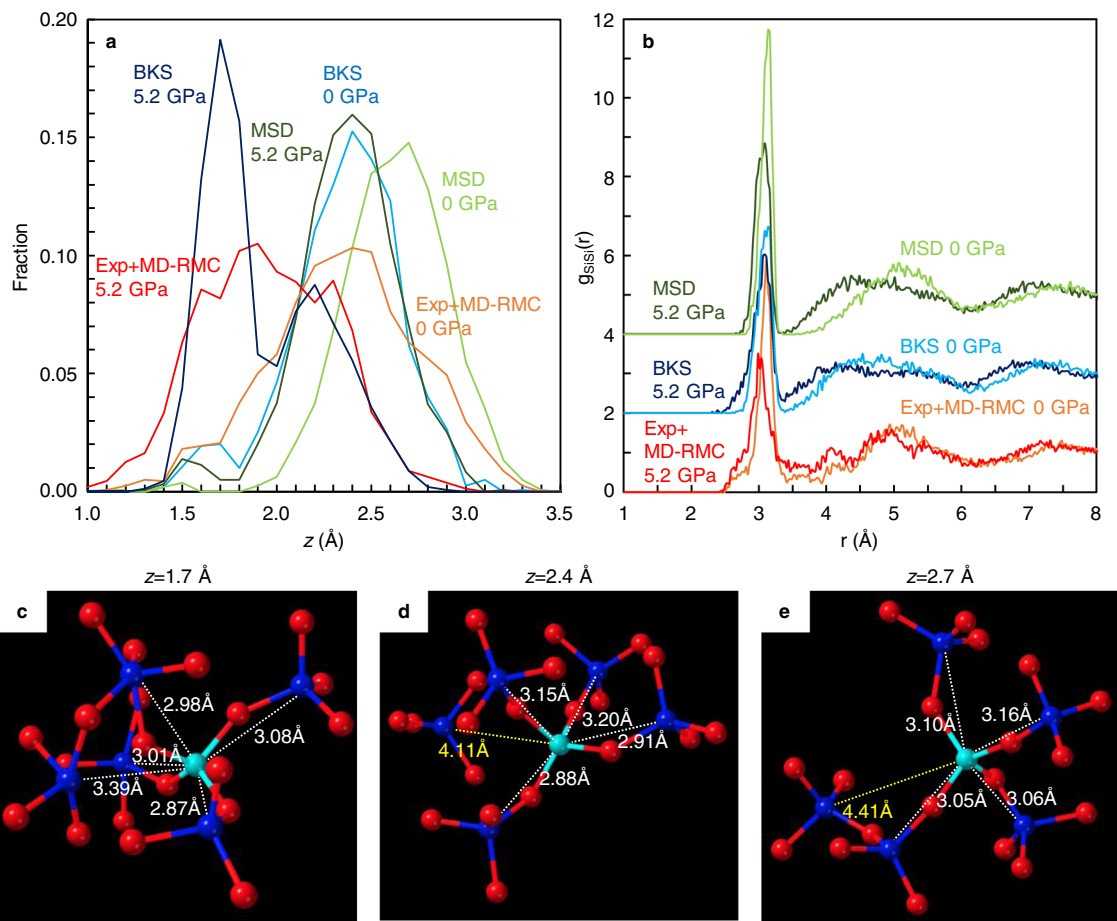

**Fig. 5 Structural features of SiO₂ glass under pressure. a** Translational order in SiO₂ glass as a function of the parameter $z$ obtained in our experiment with MD-RMC modeling (Exp+MD-RMC) and MD simulations with BKS and MSD models at 0 and 5.2 GPa. Source data are provided as a Source Data file. **b** Partial pair distribution functions of Si-Si [$g_{SiSi}(r)$]. The $g_{SiSi}(r)$ of the BKS and MSD models are displayed by a vertical offset of +2 and +4, respectively. **c–e** Structural features of SiO₂ glass with the characteristic distribution of $z = 1.7$ Å in the BKS model at 5.2 GPa (**c**), $z = 2.4$ Å in the BKS model at 0 GPa (**d**), and $z = 2.7$ Å in the MSD model at 0 GPa (**e**). Blue and light blue spheres represent silicon atoms and red spheres represent oxygen atoms. Numbers represent distances to the nearest five silicon atoms (blue spheres) from a silicon atom shown by light blue.

with a diameter of 2.5 mm and height of ~1.5 mm was placed into the MgO ring with a BN cap. Pressures were determined by the equation of the state of MgO[27]. Densities of SiO₂ glass at high-pressure conditions were calculated from experimentally measured density up to 8.6 GPa[28,29].

**Pair distribution function measurement**. Pair distribution function measurements were conducted at the beamlines BL37XU and BL05XU of SPring-8, Japan. At BL37XU, we used a monochromatic beam ($\Delta E/E = 1.8 \times 10^{-5}$) with a double crystal monochromator that employs Si511 and Si333 for the first and second reflection, respectively, at a photon energy of 40.0 keV. The X-ray beam was focused from 1 to 0.2 mm in horizontal with a 0.7-m-long horizontal-deflection mirror so as to increase an available flux. On the other hand, a high-flux pink beam ($\Delta E/E = 1.7 \times 10^{-2}$) at a photon energy of 40.3 keV was generated with a double multilayer monochromator at BL05XU[30]. The double multilayer monochromator with a wide energy bandwidth of 1.7% enhances the X-ray flux by three orders of magnitude, compared with that of a conventional beamline using a double crystal monochromator of silicon.

The structure of SiO₂ glass was measured at in situ high-pressure conditions up to 6.0 GPa in the PE cell by high-energy X-ray diffraction measurement using a CdTe point detector (Amptek X-123) with double-slit collimation setup in front of the detector. The first and second collimation slits locate at 25 and 60 cm from the sample, respectively. The double-slit collimation setup yields a collimation length of <2.5 mm at the sample position at 2θ angles higher than 5.5°, and we can obtain signals only from the SiO₂ glass sample. Sizes of incident slit and two collimation slits were adjusted with varying 2θ angle to maximize signal intensity by increasing collimation length within the diameter of the SiO₂ glass sample. High-energy X-ray diffraction measurements were carried out by scanning the 2θ angle from 1° to 60° at the beamline BL37XU and from 1° to 70° at the beamline BL05XU. Analysis was conducted by using the method developed at the beamline BL04B2 of SPring-8 (ref. [31]). We obtained the Faber–Ziman structure factor, S(Q), of SiO₂ glass at the range of the momentum transfer Q up to 19 Å⁻¹ at the beamline BL37XU and up

to 20 Å⁻¹ at the beamline BL05XU, which is almost two times larger than that in conventional high-pressure angle-dispersive X-ray diffraction measurements using monochromatic X-ray.

**MD-RMC modeling**. We determined the structural model of SiO₂ glass at high pressures, which reproduces the experimentally observed S(Q) under in situ high-pressure conditions, by using the Reverse Monte Carlo (RMC) program (RMC_POT)[32]. The structural refinement of SiO₂ glass was conducted by using an initial structure simulated by the molecular dynamics (MD) method, and the final structure model was determined by refining atomic configurations using the RMC program to reproduce the experimentally observed S(Q). This method is referred to as MD-RMC modeling, and it has been used by previous ambient pressure studies for SiO₂ glass[18]. We first constructed an initial structure using MD simulation using DL-POLY Classic[33]. About 1600 atoms of silicon and 3200 atoms of oxygen are placed in the cell at a minimal distance of 1.0 Å. Then, the structural relaxation was performed at 5000 K followed by cooling to 300 K at the rate of 0.05 K/fs. All the simulations were conducted using NVT ensemble and empirical potential[34]. After preparing the initial structures using the MD method, we refined atomic configurations by the RMC program to fit the experimentally observed S(Q). The minimum bond lengths were set at 1.35 Å for Si-O, 2.00 Å for O-O, and 2.40 Å for Si-Si to avoid unrealistic atomic configurations. The final atomic configurations, whose structure factor is consistent with the experimentally observed S(Q), was obtained after more than 2,000,000 atomic movements.

The volume of the cavity in the MD-RMC structure model of SiO₂ glass at in situ high-pressure conditions was analyzed by dividing the MD-RMC model volume into 100 × 100 × 100 cells. The size of the MD-RMC model in each direction is 41.709 Å at 0 GPa and 39.300 at 6.0 GPa, and the size of each cell for the cavity analysis is 0.41709 and 0.39300 Å, respectively. If there is no atom at the surrounding 1.35 Å distance from a cell, the cell is identified as cavity. The threshold value of 1.35 Å distance corresponds to the ionic radius of oxygen[35].

Void radius analysis by the Delaunay tetrahedralization was performed for understanding the distribution of silicon in the MD-RMC structure model. The MD-RMC model is divided into the set of tetrahedra by the tetrahedralization. Only silicon atoms are used in the tetrahedralization. A circumscribed sphere of the tetrahedron is defined as a void. Only the largest void out of the overlapped smaller voids is used for the analysis. The radius of the void illustrates the spatial distribution of silicon in the MD-RMC structure of $SiO_2$ glass under in situ high-pressure conditions.

**MD simulation**. MD simulations of $SiO_2$ glass were performed using MXDORTO code[36]. The systems are composed of 1600 atoms of silicon and 3200 atoms of oxygen. We imposed periodic boundary conditions in all directions. Ewald summations were applied to evaluate the long-range Coulombic interactions. All the atoms were moved by applying the velocity Verlet algorithm[37,38] at a time interval of 1.0 fs. The structural relaxation was performed at 4000 K with NPT ensembles for 5.0 ns, followed by cooling to 300 K by the rate of $5.0 \times 10^{11}$ K/s. The pressure is kept at 0.0001 and 5.2 GPa, respectively, to compare with experiments. The structure was relaxed at 300 K with NPT ensembles for 2.0 ns, and the trajectories of atoms were obtained under NVT ensembles at 300 K for 1.0 ns by an interval of 1.0 ps. We adopted two potential models. One is the pair potential model proposed by ref. [24] with repulsion wall[7] (BKS model). The BKS model has been used in theoretical studies of $SiO_2$ liquid[4,7]. Another is the pair and three-body potential model proposed by ref. [25] (MSD model).

## Data availability

The data that support the findings of this study are available from the corresponding author on request.

## Code availability

We used MXDORTO code for MD simulations, which is available from a co-author (Fumiya Noritake, fnoritake@yamanashi.ac.jp) upon request.

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

## Acknowledgements

This research is supported by JSPS KAKENHI (Grant Numbers: 19KK0093 and 20H00201 to Y.K.) and the SACLA/SPring-8 Basic Development Program. The experiments were performed at the BL37XU (JASRI Proposal No.: 2019B1111 and 2020A0600) and BL05XU beamlines of SPring-8.

## Author contributions

Y.K. devised the project and wrote the paper with input from all co-authors. Y.K. and K.O. developed pair distribution function measurement combined with PE press high-pressure experiment. Y.K., K.O., and N.M.K. carried out the experiments with support from K.N. and O.S. for the BL37XU beamline and from Y. Higo, Y.T., H. Yumoto, T.K., H. Yamazaki, Y.S., H.O., S.G., I.I., Y. Hayashi, K.T., T.O., J.Y., and M.Y. for BL05XU beamline. K.O., H. Yamada, and S.H. conducted the MD-RMC modeling. F.N. carried out the MD simulations. All authors discussed the results of the manuscript.

## Competing interests

The authors declare no competing interests.

## Additional information

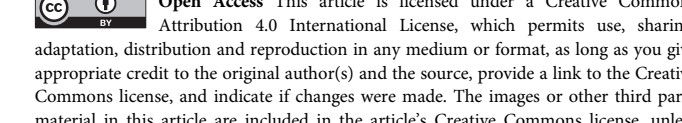

