## [Peer Review File · Nature Communications]

Experimental evidence of tetrahedral symmetry breaking in SiO₂ glass under pressureEditorial Note: This manuscript has been previously reviewed at another journal that is not operating a transparent peer review scheme. This document only contains reviewer comments and rebuttal letters for versions considered at *Nature Communications*.

REVIEWER COMMENTS

Reviewer #1 (Remarks to the Author):

The answers of the authors to my comments are only partially convincing. The description given of the structural differences between the SiO₂ polymorphs is still ambiguous. The radius of gyration parameter seems to be unappropriated to follow the proposed transition since it does not follow the density trend between the high temperature and high pressure polymorphs of Quartz. It would also have been interesting to know if the authors were considering the alfa or beta phases.

For my second comment, the limitation on the interpretation is coming from the monotonous nature of the evolution of the volume fraction of cavity. A two-state model should have present a discontinuity at least an inflection point.

As promising that could be these results, they are only partially conclusive and therefore do not constitute a significant breakthrough. I will recommend publishing this interesting work in a more appropriate journal.

Reviewer #2 (Remarks to the Author):

With the addition of figure 1b the experimental evidence is much clearer. It is important what is being claimed has information directly seen in the data without simulations. I think the authors have seriously addressed many of the concerns. I recommend the paper to be published in Nature Communication.

Review of manuscript NCom-333065_0

Experimental evidence of two distinct local structures in SiO₂ glass under pressure

Yoshio Kono and co-workers

The authors have given an answer to most of the questions and comments raised in the first round of reviews from their nature physics submission. However, they did not provide further analysis and these answers remain more on the descriptive part and give more explanation to what they have been doing, nothing really added.

The biggest problem I see is that the possible structural models come from the RMC modelling and fitting of their data and are not compared with other MD structures. I believe the study would gain a lot if they could combine MD structures and S(Q) calculation plus similar analysis with the present data set and RMC analysis. In the end they extract a correlation parameter between peaks from the S(Q) and fitted g(r) but there is no structure given from that. So it is a parameter of correlation that gives the insight of a possible structural variability. There are no real structures obtained from this study.

I understand this is may be too much to ask at this stage and it is probably someone else work to perform. I cannot say that I am very enthusiastic about this paper as I am still unconvinced that one can deduce or confirm distinct structural unit solely from the data unlike for water. Everything holds because of the correlation of the main peak at 1.6 Å and the second peak at 2.9 Å. It would be nice to have a better evaluation of the errors (based on different models from the supplementary figure 2) and how they would propagate on the structural determination. Actually, how is the structure changing from the two models discussed in this figure?

At the same time, I cannot say that I am against the publication of this study, as it is a very fine experimental work.

Response to the reviewers' comments

Response to the Reviewer #1:

Reviewer #1: The answers of the authors to my comments are only partially convincing. The description given of the structural differences between the SiO₂ polymorphs is still ambiguous. The radius of gyration parameter seems to be unappropriated to follow the proposed transition since it does not follow the density trend between the high temperature and high pressure polymorphs of Quartz. It would also have been interesting to know if the authors were considering the alfa or beta phases.

Response: We understand that the radius of gyration parameter based on SiO₂ polymorphs is not appropriate for discussing the structural change of SiO₂ glass, and therefore, in the revised manuscript, we removed the discussion based on SiO₂ polymorphs and the radius of gyration.

On the other hand, in order to clarify the structural features of SiO₂ glass under pressure, we added MD simulations at 0 GPa and 5.2 GPa, which is suggested by the reviewer #3. Similarly to the result of our experiment combined with MD-RMC modelling (Exp+MD-RMC), the MD simulation with the BKS potential model at 5.2 GPa shows clear bimodal feature in the structural parameter z (Fig. 5a), which enabled us to clarify two distinct structural units in SiO₂ glass with the characteristic distributions of z at 1.7 Å and 2.4-2.7 Å. The structure of SiO₂ glass with the characteristic distribution of z at 2.4-2.7 Å at 0 GPa shows tetrahedral symmetry structure formed by the nearest four Si atoms in the first shell, and the first and second shells are clearly separated as the fifth neighbor Si atom locates in the second shell at farther distance than 4.1 Å (Figs. 5d and 5e). On the other hand, the SiO₂ glass structure with the characteristic distribution of z at 1.7 Å at 5.2 GPa shows that the fifth neighbor Si atom locates in the first shell (Fig. 5c), which indicates collapse of the second shell onto the first shell and more than four Si neighbor atoms in the first shell of Si distribution. It is important to note that the two structural features obtained in SiO₂ glass correspond to the two-state features reported in SiO₂ liquid (Shi and Tanaka, 2018, PNAS) and water (Russo and Tanaka, 2014, Nature Comm.). For water, the low-density S state shows tetrahedral structure formed by four neighbor atoms in the first shell, and shows a clear separation between the first and second shell, since the fifth neighbor atom locates in the second shell. On the other hand, the high-density ρ state of water has the fifth neighbor atom in the first shell, which indicates collapse of the second shell

onto the first shell. The structural features of the S and ρ state of water correspond to the two structural units obtained in SiO_2 glass with the characteristic distribution of z at 2.4-2.7 Å (Figs. 5d and 5e) and 1.7 Å (Fig. 5c), respectively. It has been known in theoretical studies of the two-state model that the fraction of the S and ρ state is the controlling parameter of the anomalous properties of SiO_2 liquid and water (Tanaka, 2012, EPJ; Shi and Tanaka, 2018, PNAS). Although it is difficult to determine quantitative fraction of the S and ρ state from our results under pressure, the fraction of the ρ state structure with the distribution of z at ~ 1.7 Å markedly increase with decreasing the fraction of the S state structure above ~ 2.3 GPa (Fig. 2), which is consistent with the two-state model features suggested by the theoretical studies as the cause of the anomalous properties of SiO_2 liquid and water (Tanaka, 2012, EPJ; Shi and Tanaka, 2018, PNAS).

We added these discussions in pages 9-11.

Reviewer #1: For my second comment, the limitation on the interpretation is coming from the monotonous nature of the evolution of the volume fraction of cavity. A two-state model should have present a discontinuity at least an inflection point.

As promising that could be these results, they are only partially conclusive and therefore do not constitute a significant breakthrough. I will recommend publishing this interesting work in a more appropriate journal.

Response: Although the volume fraction of cavity shows monotonous behavior, possibly due to compensation with bulk density change, we find discontinuous change in the pressure dependence of the void space formed from Si atoms (Si void space) at ~ 2.3 -3.5 GPa (newly added Figure 4c). Figure 4c shows that Si void space anomalously shrinks more than the change of the bulk volume of SiO_2 glass below ~ 2.3 -3.5 GPa, while the results above 3.5 GPa shows almost constant ratio between the Si void space and bulk volume, indicating that the Si void space changes equally to the bulk volume change of SiO_2 glass. The data imply that anomalous properties of SiO_2 glass is related to the change of the distribution of Si atoms under pressure. It has been known in theoretical studies of the two-state model of SiO_2 liquid that the second shell structure of Si distribution is the key to characterize anomalous properties of SiO_2 liquid (Saika-Voivod et al., 2000, PRE; Shi and Tanaka, 2018, PNAS), and the fraction of the S and ρ state is the controlling parameter of the anomalous properties of SiO_2 liquid. Similarly to the theoretical studies of SiO_2 liquid, we identified the presence of the S

and ρ state structures in SiO₂ glass, as explained in the response to the above comment. The fraction of the ρ state structure with the distribution of z at ~ 1.7 Å markedly increase with decreasing the fraction of the S state structure above ~ 2.3 GPa (Fig. 2), which corresponds to the two-state model feature suggested by the theoretical studies as the cause of the anomalous properties of SiO₂ liquid. In addition, the pressure condition is consistent with that of the discontinuous change in the Si void space (Fig. 4c). These data imply correlation between the discontinuous change in the Si void space and the behavior of the S and ρ state structures in SiO₂ glass under pressure, as the cause of the anomalous properties of SiO₂ glass.

We added the description about the discontinuous change in the Si void space in pages 9.

Response to the Reviewer #2:

Reviewer #2: With the addition of figure 1b the experimental evidence is much clearer. It is important what is being claimed has information directly seen in the data without simulations. I think the authors have seriously addressed many of the concerns. I recommend the paper to be published in Nature Communication.

Response: We appreciate the reviewer's recommendation.

Response to the Reviewer #3:

Reviewer #3: The authors have given an answer to most of the questions and comments raised in the first round of reviews from their nature physics submission. However, they did not provide further analysis and these answers remain more on the descriptive part and give more explanation to what they have been doing, nothing really added.

The biggest problem I see is that the possible structural models come from the RMC modelling and fitting of their data and are not compared with other MD structures. I believe the study would gain a lot if they could combine MD structures and $S(Q)$ calculation plus similar analysis with the present data set and RMC analysis. In the end they extract a correlation parameter between peaks from the $S(Q)$ and fitted $g(r)$ but there is no structure given from that. So it is a parameter of correlation that gives the insight of a possible structural variability. There are no real structures obtained from

this study.

Response: We appreciate the suggestion by the reviewer. By following the reviewer's suggestion, we conducted MD simulations using two potential models (BKS and MSD models) at 0 GPa and 5.2 GPa, and we succeeded to clarify the two distinct structural units in SiO₂ glass under pressure, represented by the bimodal feature in the structural parameter z (Fig. 5). The structural parameter z represents the translational order of the silicon's second shell (Shi and Tanaka, 2018, PNAS). The translational order of both BKS and MSD model at 0 GPa shows a single distribution at $z=2.4$ Å and 2.7 Å, respectively, while the BKS model at 5.2 GPa shows clear bimodal distribution (Fig. 5a). The BKS model at 5.2 GPa shows significant increase of the distribution of z at 1.7 Å with decrease of the distribution of z at 2.4 Å, which is consistent with the bimodal feature obtained in our experiment with MD-RMC modelling (Exp+MD-RMC) (Fig. 5a). Based on the two distinct distributions of z at 1.7 Å and 2.4-2.7 Å, we succeeded to clarify two distinct structural units in SiO₂ glass from the MD simulations. The structure of SiO₂ glass with the distribution of z at 2.4-2.7 Å shows tetrahedral symmetry structure formed by the nearest four Si atoms in the first shell, and the first and second shells are clearly separated as the fifth neighbor Si atom locates in the second shell at farther distance than 4.1 Å (Figs. 5d and 5e). On the other hand, the structure of SiO₂ glass with the distribution of z at 1.7 Å shows that the fifth neighbor Si atom locates in the first shell (Fig. 5c), which indicates collapse of the silicon's second shell onto the first shell and more than four Si neighbor atoms in the first shell.

We added these discussions about the two distinct structural units in SiO₂ glass in pages 9-10.

Reviewer #3: I understand this is may be too much to ask at this stage and it is probably someone else work to perform. I cannot say that I am very enthusiastic about this paper as I am still unconvinced that one can deduce or confirm distinct structural unit solely from the data unlike for water. Everything holds because of the correlation of the main peak at 1.6 Å and the second peak at 2.9 Å. It would be nice to have a better evaluation of the errors (based on different models from the supplementary figure 2) and how they would propagate on the structural determination. Actually, how is the structure changing from the two models discussed in this figure?

At the same time, I cannot say that I am against the publication of this study, as it is a very fine experimental work.

Response: As we explained in the response to the above comment, in the revised manuscript, we clarified two distinct structural units in SiO₂ glass by using MD simulations (Fig. 5). It is important to note that the two structural units obtained in SiO₂ glass under pressure in this study correspond to the two-state features in SiO₂ liquid (Shi and Tanaka, 2018, PNAS) and water (Russo and Tanaka, 2014, Nature Comm.). For water, the low-density *S* state shows tetrahedral structure formed by four neighbor atoms in the first shell, and shows clear separation between the first and second shell, since the fifth neighbor atom locates in the second shell. On the other hand, the high-density ρ state of water shows the fifth neighbor atom in the first shell, which indicates collapse of the second shell onto the first shell. The structural features of the *S* and ρ state of water correspond to the two structural units in SiO₂ glass with the characteristic distribution of *z* at 2.4-2.7 Å (Figs. 5d and 5e) and 1.7 Å (Fig. 5c), respectively. These data indicate the presence of the two distinct (*S* and ρ state) structures in SiO₂ glass under pressure and the relevance of the two-state model description to SiO₂ glass as well as SiO₂ liquid and water. It has been known in theoretical studies of the two-state model of SiO₂ liquid and water (Tanaka, 2012, EPJ; Shi and Tanaka, 2018, PNAS) that the fraction of the *S* and ρ state is the controlling parameter of the anomalous properties of SiO₂ liquid and water. Although it is difficult to determine quantitative fraction of the *S* and ρ state in SiO₂ glass under pressure from our results, the fraction of the ρ state structure with the distribution of *z* at ~1.7 Å markedly increase with decreasing the fraction of the *S* state structure above ~2.3 GPa (Fig. 2), which corresponds to the two-state model feature suggested by the theoretical studies as the cause of the anomalous properties of SiO₂ liquid and water (Tanaka, 2012, EPJ; Shi and Tanaka, 2018, PNAS).

We added these discussions about the relevance of the two-state model description to SiO₂ glass in pages 10-11.

Regarding structural difference between the two MD-RMC models in Supplementary Figure 2, difference in the distribution of *z* (Supplementary Figure 2b) indicates different fraction of the *S* and ρ state structures in SiO₂ glass. The broken black line model shows lower intensity in the distribution of *z* at ~1.6-1.7 Å and higher intensity of the distribution of *z* at >~2.0 Å than the final MD-RMC model (solid black line) (Supplementary Figure 2b), which indicate underestimation of the fraction of the ρ state structure and overestimation of the fraction of the *S* state structure in the broken black line model, due to insufficient reproduction of the experimentally observed S(Q), compared to the final MD-RMC model (solid black line) (Supplementary Figure 2a).

REVIEWER COMMENTS

Reviewer #1 (Remarks to the Author):

This time the authors really took in account my comments and I thank them for that. The discussion on the silica polymorphs was removed. The new added paragraph comparing the result on SiO₂ directly to H₂O is interesting. The newly added Figure 4c is not very clear but helps to understand how much of the glass volume variation is related to the disappearance of the Si Void. Error bars would have been welcome. Other all, the points made by the authors are more convincing.

This paper can be published as it is.

Reviewer #2 (Remarks to the Author):

I am pleased with the authors response. I recommend publication.

Reviewer #3 (Remarks to the Author):

Experimental evidence of two distinct local structures in SiO₂ glass under pressure
Yoshio Kono and co-workers

The authors made quite some effort to include some MD calculations in this last version, and it is very much appreciated although it seems to show slightly different results at high pressure at least from the S(Q) data.

I still have some reserves about the parallel that is made between SiO₂ glass to liquid water model. The first parallel that is found in literature is mostly between T dependence in liquid water and liquid silica and there is no experimental evidence for this parallel. Nor for the pressure dependence for both (only water has been investigated). Here the authors go one step beyond with the comparison of liquid water and Silica glass (as the frozen melt). It seems that for water the collapse of the second shell is due to the breaking of hydrogen bonds and the collapse of the distance of the second shell away to the first one. For the glass this would mean breaking covalent bonds and I have a problem to understand this, especially that the glass shows a pure elastic response in this pressure regime, coming back to its original density and structure.

As I mentioned in a previous revision, the parallel between silica glass and SiO₂ liquid is not demonstrated and it is known that silica glass has an open structure at ambient conditions that may not reflect the structure of the real frozen liquid (this maybe the case for more depolymerized glasses) as it is demonstrated by its anomalous low density. Thus, I am worried that the effect described here may be an effect of the void collapse from the original structure rather than something else. Could it be that because the experiment is carried without pressure medium, the changes that are observed may not be from the network itself but rather a compaction mechanism of the void and loss of the porosity?

Lastly, I don't really grasp the impact and consequence of the work presented here. It is definitely a fine experimental work with a lot of effort to build the model but there are no real

implications discussed apart from the fact that this may mean a similar behaviour than for water. But again comparing a van-der-walls liquid with a covalent solid is a stretch. Although the paper has quite some merit on the experimental side, I have too much doubt at this point to consider a clear evidence of a double structure distribution in SiO₂ glass with pressure.

Response to the reviewers' comments

Response to the Reviewer #1:

Reviewer #1: This time the authors really took in account my comments and I thank them for that. The discussion on the silica polymorphs was removed. The new added paragraph comparing the result on SiO₂ directly to H₂O is interesting. The newly added Figure 4c is not very clear but helps to understand how much of the glass volume variation is related to the disappearance of the Si Void. Error bars would have been welcome. Other all, the points made by the authors are more convincing.

This paper can be published as it is.

Response: We appreciate the reviewer's recommendation.

Response to the Reviewer #2:

Reviewer #2: I am pleased with the authors response. I recommend publication.

Response: We appreciate the reviewer's recommendation.

Response to the Reviewer #3:

Reviewer #3: The authors made quite some effort to include some MD calculations in this last version, and it is very much appreciated although it seems to show slightly different results at high pressure at least from the S(Q) data.

I still have some reserves about the parallel that is made between SiO₂ glass to liquid water model. The first parallel that is found in literature is mostly between T dependence in liquid water and liquid silica and there is no experimental evidence for this parallel. Nor for the pressure dependence for both (only water has been investigated). Here the authors go one step beyond with the comparison of liquid water and Silica glass (as the frozen melt). It seems that for water the collapse of the second shell is due to the breaking of hydrogen bonds

and the collapse of the distance of the second shell away to the first one. For the glass this would mean breaking covalent bonds and I have a problem to understand this, especially that the glass shows a pure elastic response in this pressure regime, coming back to its original density and structure.

Response: By considering the reviewer's comment, the editor of Nature Communications suggested removing the claims about connection with the structure of liquid water. We therefore removed all the descriptions about the comparisons between water and SiO₂ glass in the revised manuscript by following the editor's suggestion.

Regarding investigation about pressure dependence, theoretical study of SiO₂ liquid by Shi and Tanaka (2018) has reported the change of the fraction of the low-density *S* state not only as a function of temperature at ambient pressure but also as a function of pressure at 5,000 K. To introduce the study of Shi and Tanaka (2018) about both temperature and pressure dependences, we added following sentence at lines 11-15 of page 9:

The theoretical study of SiO₂ liquid⁴ assigned the *S* and ρ states to the high and low distributions in the parameter z , respectively, and showed that the fraction of the low-density *S* state is the controlling parameter of the anomalous density behavior of SiO₂ liquid not only as a function of temperature at ambient pressure but also as a function of pressure at 5,000 K.

In addition, we consider that covalent bond does not break in this pressure range and structure of SiO₂ glass comes back to original structure. A previous study of Onodera et al. (2020) (Ref.17) reported ambient pressure analysis of the structure of SiO₂ glass synthesized at 7.7 GPa and room temperature. The result shows a single distribution in the parameter z at $\sim 2.4-2.5$ Å without peak at $z=1.6-1.7$ Å, which corresponds to the *S* state structure with tetrahedral symmetry obtained at 0 GPa in this study. The data imply that the collapse of the silicon's second shell onto the first shell observed in SiO₂ glass under in situ pressure condition comes back to original tetrahedral symmetry structure when the sample is recovered at ambient pressure.

Reviewer #3: As I mentioned in a previous revision, the parallel between silica glass and SiO₂ liquid is not demonstrated and it is known that silica glass has an open structure at ambient conditions that may not reflect the structure of the real frozen liquid (this maybe the case for more depolymerized glasses) as it is demonstrated by its anomalous low density.

Thus, I am worried that the effect described here may be an effect of the void collapse from the original structure rather than something else. Could it be that because the experiment is carried without pressure medium, the changes that are observed may not be from the network itself but rather a compaction mechanism of the void and loss of the porosity?

Response: We added following sentences to explain the similarity of the structural features between SiO₂ glass and SiO₂ liquid in pages 10-11:

It is important to note that the two structural features obtained in SiO₂ glass under pressure in this study correspond to the *S* and ρ state structures of SiO₂ liquid reported in the theoretical study⁴. For SiO₂ liquid, the low-density *S* state has four silicon neighbor atoms in the first shell, and exhibits high tetrahedral order with clear separation between the first and second shell⁴. On the other hand, the high-density ρ state of SiO₂ liquid shows more than four silicon atoms in the first shell by collapsing the second shell onto the first shell, and exhibits low tetrahedral order. The structural features of the *S* and ρ state of SiO₂ liquid correspond to the two structural features obtained in SiO₂ glass with the characteristic distribution of *z* at 2.4-2.7 Å (Figs. 5d and 5e) and 1.7 Å (Fig. 5c), respectively. These observations indicate the similarity of the structural behavior in SiO₂ glass under pressure obtained in this study to that theoretically simulated in SiO₂ liquid at high temperatures and high pressures⁴. SiO₂ glass mainly consists of the low-density *S* state with tetrahedral symmetry structure at low pressures. On the other hand, the local tetrahedral symmetry structure breaks at high pressures, and the fraction of the low-density *S* state in SiO₂ glass decreases under pressure, as well as theoretical observation in SiO₂ liquid at high temperatures and high pressures⁴.

In addition, we have shown volume fraction of cavity as a function of pressure in Fig. 4b, which represents change of porosity in SiO₂ glass with varying pressure. The result shows monotonous decrease of the volume fraction of cavity with increasing pressure. The fact indicates that there is no marked change in the pressure dependence of porosity up to 6.0 GPa, and therefore loss of porosity is not related to our observed structural behavior in SiO₂ glass under pressure.

Reviewer #3: Lastly, I don't really grasp the impact and consequence of the work presented

here. It is definitely a fine experimental work with a lot of effort to build the model but there are no real implications discussed apart from the fact that this may mean a similar behaviour than for water. But again comparing a van-der-walls liquid with a covalent solid is a stretch. Although the paper has quite some merit on the experimental side, I have too much doubt at this point to consider a clear evidence of a double structure distribution in SiO₂ glass with pressure.

Response: By considering the reviewer's comments, the editor of Nature Communications suggested removing the claims about the existence of two local structures and connection with the structure of liquid water. By following the editor's suggestion, we removed all the descriptions about the comparisons between water and SiO₂ glass, and rephased the existence of two local structures into breaking of tetrahedral symmetry structure. Theoretical studies of SiO₂ liquid have suggested that the second shell structure of silicon is the key to understand the anomalous properties at high temperatures and high pressures, and Shi and Tanaka (2018) reported that the fraction of the low-density *S* state with tetrahedral symmetry structure is the controlling parameter of the anomalous density behavior of SiO₂ liquid. However, it has not been identified in experiment. This paper reports the first experimental finding of a bimodal behavior in the translational order of silicon's second shell in SiO₂ glass under pressure, and breaking of tetrahedral symmetry structure (*S* state structure) in SiO₂ glass at high pressures, which correspond to theoretical observation in SiO₂ liquid at high temperatures and high pressures as structural origin of the anomalous properties.

Understanding the structural origin of the anomalous properties of tetrahedral liquids and amorphous materials at high temperature and high pressure conditions is of great interest in wide range of scientific fields such as physics, chemistry, geoscience, and materials science. Our study opens new way to experimentally understand structural behavior of SiO₂ glass under pressure, as well as other tetrahedral liquids and amorphous materials.